# Adversarial Imitation Learning with Preferences

**Aleksandar Taranovic**[1,2][*]**, Andras Kupcsik**[2]**, Niklas Freymuth**[1]**, Gerhard Neumann**[1]
[1] Autonomous Learning Robots Lab, Karlsruhe Institute of Technology, Karlsruhe, Germany
[2] Bosch Center for Artificial Intelligence, Renningen, Germany

## Abstract

Designing an accurate and explainable reward function for many Reinforcement Learning tasks is a cumbersome and tedious process. Instead, learning policies directly from the feedback of human teachers naturally integrates human domain knowledge into the policy optimization process. Different feedback modalities, such as demonstrations and preferences, provide distinct benefits and disadvantages. For example, demonstrations convey a lot of information about the task but are often hard or costly to obtain from real experts while preferences typically contain less information but are in most cases cheap to generate. However, existing methods centered around human feedback mostly focus on a single teaching modality, causing them to miss out on important training data while making them less intuitive to use. In this paper we propose a novel method for policy learning that incorporates two different feedback types, namely *demonstrations* and *preferences*. To this end, we make use of the connection between discriminator training and density ratio estimation to incorporate preferences into the popular Adversarial Imitation Learning paradigm. This insight allows us to express loss functions over both demonstrations and preferences in a unified framework. Besides expert demonstrations, we are also able to learn from imperfect ones and combine them with preferences to achieve improved task performance. We experimentally validate the effectiveness of combining both preferences and demonstrations on common benchmarks and also show that our method can efficiently learn challenging robot manipulation tasks.

## 1 Introduction

This paper aims to progress research towards enabling humans without expert knowledge of machine learning or robotics to teach robots to perform tasks using diverse feedback modalities. Enabling human teachers to use various feedback types allows for a more natural human-robot training interaction. In particular, this paper focuses on human *demonstrations* of the desired behaviour, and *preferences*, which are pairwise comparisons of two possible robot behaviors. Both types of feedback have distinct benefits. Demonstrations exploit the domain knowledge of human teachers for a given task. Yet, they are often cumbersome to generate, and present considerable cognitive load on the human teacher, up to the point that the teacher might not be capable to demonstrate optimal behavior on their own. In contrast, preferences are easier to evaluate, albeit less informative as they only indicate relative quality, i.e., which option is better among two provided ones.

Both feedback types have been extensively researched in isolation. For instance, a plethora of approaches focus on learning from demonstrations, also referred to as imitation learning (Osa et al., 2018). In recent years, preference learning has been an active research topic (Wirth et al., 2017), but integrative work on learning from multiple feedback modalities has been much sparser and is often limited to e.g., a theoretical analysis (Jeon et al., 2020).

In this paper, we introduce Adversarial Imitation Learning with Preferences (AILP), a novel method for learning from a combination of demonstrations and preferences that builds upon the well-known Adversarial Imitation Learning (AIL) framework. AIL uses a discriminator that indicates the change

---

[*]Corresponding author: aleksandar.taranovic@partner.kit.edu

of the policy, which can be seen as a differential reward. In preference learning, typically we directly encode the reward in a preference and require a static reward. We show that combining a differential and static reward directly in the form of adversarial comparisons and preferences is incompatible and leads to poor performance. To alleviate this issue, we present a preference loss that is compatible with AIL and therefore enables AIL approaches to benefit from additional preference feedback that is available alongside demonstrations. We provide an overview of our method in Figure 1.

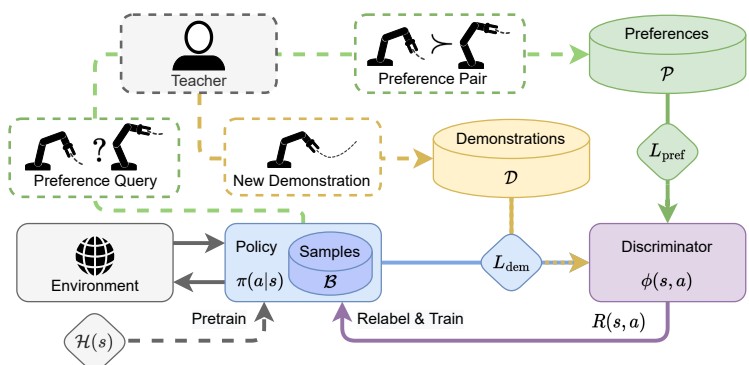

Figure 1: Schematic overview of Adversarial Imitation Learning with Preferences (AILP). Given a policy (blue) that may be pre-trained on a self-supervised maximum entropy objective, we can optionally query a teacher for new demonstrations (yellow) and preferences between trajectories from the buffer $\mathcal{B}$ (green). Next, a discriminator $\phi(s, a)$ (purple) is trained to discriminate between samples from $\mathcal{B}$ and teacher demonstrations ($L_{\text{dem}}$), while at the same time preferring better samples over worse ones ($L_{\text{pref}}$). For an off-policy Reinforcement Learning (RL) algorithm, this training makes use of the environment (grey) and a sample buffer that needs to be relabeled according to the reward that depends on $\phi(s, a)$. This process is iterated over until convergence, at which point the policy produces samples that are indistinguishable from the expert demonstrations.

The contribution of the paper are as follows: (i) we present a novel preference-learning approach that is compatible with Adversarial Imitation Learning and achieves results comparable to the state of the art, (ii) we extend Adversarial Imitation Learning to include learning from both preferences and demonstrations, (iii) we present extensive evaluations of the proposed method, outperforming various baseline methods on well-known benchmarks.

## 2 RELATED WORK

Teaching robots by human teachers has been an active research topic in the past years (Chernova & Thomaz, 2014). There has been research on unifying multiple feedback types into a single framework, e.g., by (Jeon et al., 2020), but the focus has been mostly on *unimodal* feedback. Two commonly researched human feedback modes are *demonstrations* and *preferences*. Besides these, there are numerous others, such as assigning numerical values to different options by human teachers, as in the work by Wilde et al. (2021).

**Learning from Demonstrations.** Recently, learning from demonstrations and Imitation Learning (IL) (Osa et al., 2018; Argall et al., 2009) have been active research fields in robot learning in which the robot is presented with instances of desired expert behavior. IL algorithms can generally be classified as either Behavioral Cloning (BC) (Torabi et al., 2018; Florence et al., 2021), where a policy is directly regressed from demonstrations in a supervised fashion, or as Inverse Reinforcement Learning (IRL) (Abbeel & Ng, 2017; Ziebart et al., 2008; Zakka et al., 2021), which recovers and subsequently optimizes a reward function from demonstrations. In recent years, there has been a rise in adversarial methods inspired by Generative Adversarial Networks (Goodfellow et al., 2014). Starting with Generative Adversarial Imitation Learning (GAIL) (Ho & Ermon, 2016), these methods utilize a discriminator-based distribution matching objective for both BC (Ho & Ermon, 2016; Torabi et al.) and IRL (Fu et al., 2018; Xiao et al., 2019). Building on this, another body of work

utilizes the inherent connection between discriminator training and (log) density ratio estimation for AIL with Gaussian Mixture Model (GMM) policies (Becker et al., 2020; Freymuth et al., 2021; 2022)

**Learning from Preferences.** Preference evaluation indicates the preferred behavior out of two possible options. Compared to demonstrations, preferences are generally easier to provide for human teachers, but contain significantly less information. Here, early work made use of Gaussian Process (GP) for encoding preferences for classification tasks (Chu & Ghahramani, 2005; Houlsby et al., 2012). Bıyık et al. (2020) have also used GPs to learn a reward function with a predefined set of behaviors. Moreover, these models can be used with trajectory features to lower the dimensionality of the problem (Sadigh et al., 2017; Myers et al., 2021; Bıyık & Sadigh, 2018). Christiano et al. (2017) recover the full policy by using a deep RL approach. An extension of this work, named Pebble (Lee et al., 2021a), combines unsupervised entropy-based pre-training with a Soft Actor-Critic (SAC) (Haarnoja et al., 2018) policy to optimize the learned reward function. Lee et al. (2021b) evaluate Pebble using different teacher profiles. In this paper we compare our method with the results presented in Lee et al. (2021a), leaving the evaluation with different teachers for future work.

**Learning from Demonstrations and Preferences.** Jeon et al. (2020) define a formalism that enables integrating various feedback types into a unified framework. However, this work is a theoretical analysis and requires further empirical evaluation. Practical methods that combine demonstrations and preferences have so far mostly been extensions of existing work for learning from one of those feedback types. In Bıyık et al. (2021), the authors propose a Bayesian method which infers an initial belief about the unknown reward function from demonstrations, and then uses preferences (similar to Bıyık & Sadigh (2018)) to refine this function to better match the true reward. Yet, this method requires trajectory-wise features, while AILP directly acts on state-action pairs. Moreover, Ibarz et al. (2018) propose bootstrapping the initial policy using demonstrations and then apply a preference learning approach (Christiano et al., 2017). In our method we integrate both demonstrations and preferences into the learning process, without any assumptions on when and what type of feedback is required for learning. Brown et al. (2019) extended work from Brown et al. and introduced a method that uses suboptimal demonstrations to train model train using behavior cloning. By using this model with different levels of noise they can self generate preferences which they learn using the same loss function as in Lee et al. (2021a).

## 3 PRELIMINARIES

In this section, we present an overview of used notations in the paper, as well as preliminaries on generative adversarial imitation learning.

**Demonstrations and preferences.** We define $T$ consecutive pairs of states $s$ and actions $a$ as a trajectory $\xi = \{s_1, a_1, ..., s_T\}$. We also interchangeably use the term rollout to refer to a trajectory. Besides full trajectories, we also consider trajectory segments, which are shorter sections of the full trajectory with length $T_{\text{seg}} < T$. A trajectory that is generated by a teacher is also referred to as a *demonstration*. Unless stated otherwise, we assume that the teacher provides optimal examples of a given behavior. *Preferences* are labeled pairs of trajectories $(\xi_1, \xi_2)$, such that $\xi_1$ is preferred over $\xi_2$. In contrast to demonstrations, where the teacher is responsible for generating the trajectory, trajectories in a preference pair are generated by the agent. We assume that the preferred trajectory has higher likelihood under the expert policy $\pi^*$ (Wirth et al., 2017), i.e.,

$$\xi_1 \succ \xi_2 \Leftrightarrow p\left(\xi_1|\pi^*\right) > p\left(\xi_2|\pi^*\right),$$

which also implies that the cumulative reward of trajectory $\xi_1$ is greater than the reward of $\xi_2$, that is,

$$R(\xi_1) > R(\xi_2), \ R(\xi) = \sum_{t=0}^{T} r(s_t, a_t),$$

where $r(s, a)$ is a reward function that is unknown to the agent.

In the literature, such as by Lee et al. (2021a), a common approach for defining the loss function is based on Bradley-Terry model (Bradley & Terry, 1952) where they model

$$p_{BT}(\xi_1 \succ \xi_2) = \frac{\exp \sum_{t=0}^{T} r(s_t^1, a_t^1)}{\sum_{i=1}^{2} \exp \sum_{t=0}^{T} r(s_t^i, a_t^i)}.$$

Based of the previous preference probability and assuming that the preferred trajectory is always labeled as $\xi_1$, the following loss function is used

$$L_{BT} = - \underset{(\xi_1,\xi_2)\sim\mathcal{P}}{\mathbb{E}} \left[\log p_{BT}(\xi_1 \succ \xi_2)\right].$$

**Preference query and labels.** Preferences are obtained by sampling trajectories from the policy, or alternatively, they can also be sampled from a buffer of existing trajectories. Lee et al. (2021b) analyze different preference querying methods in order to obtain more informative pairs. The approach that we consider in this paper is based on cross-entropy. We select a pair of trajectories from the buffer that has the largest cross-entropy value. After the agent generates the query, it is presented to the teacher for evaluation. The teacher then labels one trajectory as better according to its evaluation criteria. In simulated tasks, the teacher is often referred to as an oracle and it has access to the true underlying reward function of the task.

**Trajectory likelihood.** Assuming Markovian dynamics, the likelihood of the trajectory under the expert policy is

$$p(\xi|\pi^*) = p(s_0) \prod_{t=0}^{T-1} p(s_{t+1}|a_t, s_t)\pi^*(a_t|s_t). \tag{1}$$

**Adversarial Imitation Learning.** The main training loop of AIL revolves around iteratively training a discriminator $D$ on samples, and training a generator, which is usually given as a policy $\tilde{\pi}(a|s)$ on some reward induced by this discriminator. The discriminator and the policy are updated sequentially in a loop after minimizing their respective losses as described below.

At the start of iteration $k$, we generate a set of rollouts $\mathcal{M}_k = \{\xi\}$ following our policy $\tilde{\pi}_k(a|s)$. Using $\mathcal{M}_k$ and expert demonstrations $\mathcal{D} = \{\xi_d\}$ that are provided by an expert teacher $\pi^*(a|s)$, we update the discriminator $D(s,a) : \mathcal{A} \times \mathcal{S} \mapsto \mathbb{R}$ such that it distinguishes between expert demonstrations $(s,a) \in \mathcal{D}$ and samples $(s,a) \in \mathcal{M}_k$. After training the discriminator, it can be used to recover an intermediate reward function $R_k(s,a)$, which is subsequently used to obtain a new policy $\tilde{\pi}_{k+1}(a|s)$ using a regular Reinforcement Learning algorithm, such as Soft Actor-Critic (SAC) (Haarnoja et al., 2018). The exact form of $R_k(s,a)$ varies depending on the concrete AIL algorithm. Popular choices include $R^{\text{GAIL}}(a, s) = -\log(1 - D(a, s))$ from GAIL (Ho & Ermon, 2016) and $R^{\text{AIRL}}(a, s) = \log(D(a, s)) - \log(1 - D(a, s))$ from AIRL (Fu et al., 2018). For more details we refer to Orsini et al. (2021). The former reward optimizes the Jensen-Shannon divergence between the $\tilde{\pi}_i(a|s)$ and $\pi^*(a|s)$, while the latter optimizes the Kullback–Leibler divergence between the same distributions. After updating the policy, the above process is repeated. The algorithm converges when the policy creates samples that are indistinguishable from that of the teacher. At this point, the policy successfully imitates the behavior of the expert teacher.

## 4 ADVERSARIAL IMITATION LEARNING WITH PREFERENCES

Our approach has two main steps. In the first step, instead of a discriminator, we update a log density ratio estimator (LDRE) using the demonstration and preference loss functions that we introduce in Section 4.3. In the second step, we update our policy using the LDRE based reward and SAC. Every $K$ iterations, we add $M$ new preferences to our preference buffer. We summarize the algorithm in Section 4.5.

### 4.1 REFORMULATING AIL USING DENSITY RATIO ESTIMATION

AIL can be derived under a divergence minimization perspective such as the Jensen-Shannon Divergence (Ho & Ermon, 2016; Orsini et al., 2021), KL-divergence (Fu et al., 2018; Becker et al., 2020) or more general f-divergences (Ghasemipour et al., 2020). We will follow a KL-minimization perspective as it allows a clear interpretation of the learned discriminator (Arenz & Neumann, 2020). Similar to Fu et al. (2018); Becker et al. (2020), we start by minimizing the KL-divergence between the generator's and the demonstrator's trajectory distributions, i.e.

$$\pi^* = \text{argmin}_\pi \text{KL}\big(p^\pi(\xi)||p^q(\xi)\big), \text{ with } p^\pi(\xi) = p(s_0) \prod_{t=0}^{T-1} p(s_{t+1}|s_t, a_t)\pi(a_t|s_t). \tag{2}$$

The distribution $p^q(\xi)$ is defined similarly with unknown demonstrator policy $q(a|s)$. As demonstrator and generator trajectories use the same system dynamics, the KL minimization simplifies to

$$\text{KL}\big(p^\pi(\xi)||p^q(\xi)\big) = \int p^\pi(\xi) \sum_{t=1}^{T} \log \frac{\pi(a_t|s_t)}{q(a_t|s_t)} d\xi \tag{3}$$

$$= \sum_{t=1}^{T} \iint p^\pi(s_t)\pi(a_t|s_t) \log \frac{\pi(a_t|s_t)}{q(a_t|s_t)} ds_t da_t, \tag{4}$$

where $p^\pi(s_t)$ is the state distribution of policy $\pi$ at time step $t$. We can see that this objective depends on the log density ratio $\log \frac{\pi(a_t|s_t)}{q(a_t|s_t)}$ of the generator and demonstrator action distributions. This log density ratio can be estimated by a discriminator which is typically trained by a BCE classification loss Becker et al. (2020); Arenz & Neumann (2020). Yet, in order to break the dependency of the discriminator on the generation policy $\pi$ and following Arenz & Neumann (2020), we introduce the following trick

$$\text{KL}\big(p^\pi(\xi)||p^q(\xi)\big) = \sum_{t=1}^{T} \iint p^\pi(s_t) \left( \pi(a_t|s_t) \log \frac{\pi_k(a_t|s_t)}{q(a_t|s_t)} + \log \frac{\pi(a_t|s_t)}{\pi_k(a_t|s_t)} \right) ds_t da_t, \tag{5}$$

where $\pi_k$ is the old generator used for sampling the data. In the following, we will replace the log density ratio $\log \pi_k(a_t|s_t) - \log q(a_t|s_t)$ by an estimator $\phi(s,a)$ which is trained using the BCE objective. We further rewrite our minimization problem into an maximization problem and arrive at

$$J^\pi = \sum_{t=1}^{T} \mathbb{E}_{p^\pi(s_t)\pi(a_t|s_t)} \Big[ -\phi(s_t, a_t) + \log \pi_k(a_t|s_t) \Big] + H\big(\pi(\cdot|s_t)\big), \tag{6}$$

where $H\big(\pi(\cdot|s_t)\big)$ is the conditional entropy of policy $\pi$. Note that $J^\pi$ now needs to be maximized w.r.t $\pi$.

## 4.2 Rewards in AIL

While in standard AIL we typically use the discriminator as reward signal for updating the generator, it is important to note that the discriminator does not constitute a reward signal in the classical sense. It is non-stationary and heavily depends on the current generator policy $\pi_k$. It provides information which trajectories should be reinforced given the current generator, i.e., policy $\pi_k$ but can not be used to evaluate the quality of trajectories independently of the currently used generator. To illustrate this issue, consider the case that our model policy is perfectly matching our expert, and that the discriminator has been trained accordingly. In this case, the discriminator outputs $0.5$ for every possible input, thus being incapable of providing a metric to compare different trajectories.

Yet, a different reward formulation can be extracted from our discussion on density ratio estimation in AIL. Eq. 6 can be interpreted as average reward reinforcement learning problem with rewards

$$r_t(s_t, a_t) = \log \pi_k(a_t|s_t) - \phi(s_t, a_t) \tag{7}$$

and an additional max-entropy objective for the policy. It is easy to see that, for a perfect density ratio estimator, i.e. $\phi(s_t, a_t) = \log \pi_k(a_t|s_t) - \log q(a_t|s_t)$, the reward function would reduce to $\log q(a_t|s_t)$, breaking the dependency of the reward function definition in AIL from the generator $\pi_k(a_t|s_t)$. Hence, we can now directly use our reward definition to evaluate trajectories which will be important in Section 4.3 for introducing an additional preference loss.

## 4.3 Combining Preference and Demonstration Loss

Following our discussion from above, the demonstration loss for training the discriminator is specified by the BCE loss which results in reliable density ratio estimators, i.e.,

$$L_{dem}(\phi, \mathcal{D}, \mathcal{M}_0) = - \mathbb{E}_{(a,s)\sim\mathcal{D}} \Big[ \log \sigma\big(\phi(a,s)\big) \Big] - \mathbb{E}_{(a,s)\sim\mathcal{M}} \Big[ \log \big( 1 - \sigma(\phi(a,s)) \big) \Big], \tag{8}$$

where $\mathcal{M}$ contains samples generated by the current model policy $\tilde{\pi}_k$.

Furthermore, we need to include a loss function for the available preferences to obtain a more accurate discriminator. To this end, we will use our reward function defined in Eq. 7 to estimate the trajectory return

$$R(\xi) = \sum_t r(s_t, a_t) = \sum_t \log \pi_k(a_t|s_t) - \phi(s_t, a_t),$$

of the two trajectories $\xi_1$ and $\xi_2$ that constitute the preference pair. We now further restrict our discriminator to also comply with the given preferences using a hinge loss per preference pair,

$$L_{pref}(R, \mathcal{P}) = \mathop{\mathbb{E}}_{(\xi_1, \xi_2) \sim \mathcal{P}} \left[ \text{hinge} \left( R(\xi_1) - R(\xi_2) \right) \right], \tag{9}$$

i.e., there is an additional penalty if the estimated return of the preferred trajectory $R(\xi_1)$ is smaller then the return of $R(\xi_2)$. In our experiments, we also present an ablation study testing different loss functions for the preferences, including without addition of the log-density of the old generator in our reward definition as well as the typically used sigmoid loss for the preference pairs. We observed that the hinge loss resulted in the most stable results. Our interpretation is that this is due to the property of the hinge loss that it is only active if the preference is violated while the sigmoid loss is forcing the returns of two preference pairs to be far apart, causing instabilities.

Preference and demonstration feedback may impact the learnt policy to different degrees depending on the teacher and the task. We can compensate for this effect by increasing the influence of one feedback type by scaling the corresponding loss function. For example, we can multiply the preference loss $L_{pref}$ with a hyperparameter $\alpha$ to do so. In the case when we are only considering preferences, our method represents a novel preference-learning method.

## 4.4 OPTIMIZING THE GENERATOR

Although there are algorithms for solving average reward RL problems (Zhang & Ross, 2021), we will use standard discounted RL formulations such as SAC (Haarnoja et al., 2018) for our policy update as such off-policy methods are known to be more data efficient. While our derivations only hold for the average reward case, we do not expect this to make a major difference for larger discount factors. In this paper we concentrate on the addition of preferences in the discriminator loss and leave the usage of average reward RL algorithms for AIL for future work. We use SAC to optimize the reward, which is an off-policy method that uses a replay buffer $B$. However, because our reward definition changes with $\phi(a, s)$ we need to also relabel the data in the buffer accordingly.

As we use Gaussian policy class, the term $\log \pi_k(a_t|s_t)$ used in the reward definition was also too restrictive for SAC as policy optimizer as this term quickly tends to dominate, making the value function approximation for the first part of the reward unreliable. Hence, for policy optimization, we directly used $-\phi(s, a)$ as reward signal. Note that the resulting average reward RL problem from Eq. 6 can also be reformulated as

$$J^\pi = \sum_{t=1}^{T} \mathbb{E}_{p^\pi(s_t)\pi(a_t|s_t)} \left[ -\phi(s_t, a_t) \right] + \mathbb{E}_{p^\pi(s_t)} \left[ \text{KL}\big( \pi(\cdot|s_t) || \pi_k(\cdot|s_t) \big) \right], \tag{10}$$

i.e., we get an additional KL regularization term that punishes the derivation to the old policy. While there are several on-policy RL algorithms that directly use such KL regularization terms for their policy update (Schulman et al., 2015; 2017), we resort to off-policy actor critic methods such as SAC. In SAC, we still have the maximum entropy formulation, but we ignored the KL term. As the KL-term is a 'moving' objective that prevents the policy from making too large updates, this assumption should in theory only affect the learning speed, not the quality of the found policy. The resulting optimization corresponds to the optimization performed by AIRL (Fu et al., 2018), as presented in Orsini et al. (2021).

## 4.5 RESULTING ALGORITHM

The input to the algorithm is a set of demonstrations $\mathcal{D}$ and a set of preferences $\mathcal{P}$, which may be empty sets as well. In order to have more informative trajectories for preference evaluations already at the beginning of the learning, we pretrain the policy $\tilde{\pi}$ to maximize state entropy $\mathcal{H}(s) =$

$-\mathbb{E}\{\log p(s)\}$, as in Lee et al. (2021a). Without this step the initial trajectories would be completely random, and comparison among them is less insightful.

In each iteration of the main learning loop, we first collect additional demonstrations and request preference queries, and update the feedback sets $\mathcal{D}$ and $\mathcal{P}$. Note that this is an *optional* step, as we can update the generator policy using only the existing sets $\mathcal{D}$ and $\mathcal{P}$. To generate significantly different trajectories for preference evaluation, we first sample a batch of trajectory pairs $\{(\xi_{j,1} \succ \xi_{j,2})\}_j$ following the policy $\tilde{\pi}$. From these we select the pair that has the largest entropy $\mathcal{H}(P(\xi_{j,1} \succ \xi_{j,2}))$, where $P(\xi_{j,1} \succ \xi_{j,2}) = \mathrm{softmax}(R(\xi_{j,1}), R(\xi_{j,2}))$, as in Lee et al. (2021a). The $R(\xi)$ is an estimated reward function, described in more detail below. We present the trajectory pair to an expert to obtain the preference feedback. The expert uses its criteria to compare and label the two options. In simulations we use an oracle that labels preferences according to the defined environmental reward of the trajectories. The reward values of these trajectories are never directly provided to AILP. In the next stage we update the LDRE $\phi$. We then collect the samples from the replay buffer $B$ and afterwards, we compute the demonstration loss $L_{dem}$ and the preference loss $L_{pref}$ and optimize them w.r.t. $\phi$. We use the optimized LDRE to define an expert reward $R_{i+1}^{AIL}$ and optimize the generator $\tilde{\pi}$ using SAC. An overview of this procedure is shown in Figure 1, and we additionally provide pseudocode in Appendix B.

## 5    EXPERIMENTAL EVALUATION

With our experiments we would like to answer the following questions **Q1:** How does AILP compare with the current state of the art in preference learning? **Q2:** What is the influence of including expert demonstrations on improving policy learning in addition to preferences? **Q3:** What is the influence of using imperfect demonstrations on policy learning in addition to preferences?

We consider 6 different manipulation tasks from the metaworld benchmark (Yu et al., 2019), namely button press, door open, door close, window open, window close, drawer close. For comparison with other work we consider episodes of 250 rather than 500 steps, which is still sufficient to finish the task successfully. For evaluating the performance we consider normalized trajectory returns as well as task success rate. The success rate is 1 when a task-specific success criteria is achieved, and 0 otherwise. We report the average success rate over 10 rollouts. For computing the normalized trajectory reward, we sum all the environmental rewards of individual steps and then normalize by dividing the resulting sum with the average trajectory reward of the expert. For this metric we also consider the same 10 rollouts and we calculate their mean and standard deviation. In all experiments we use the same set of 10 random seeds. For each rollout from a task the position of the object in the scene is randomly set, e.g., location of the button in the button press task. For more details about different environments we refer to the Appendix A.

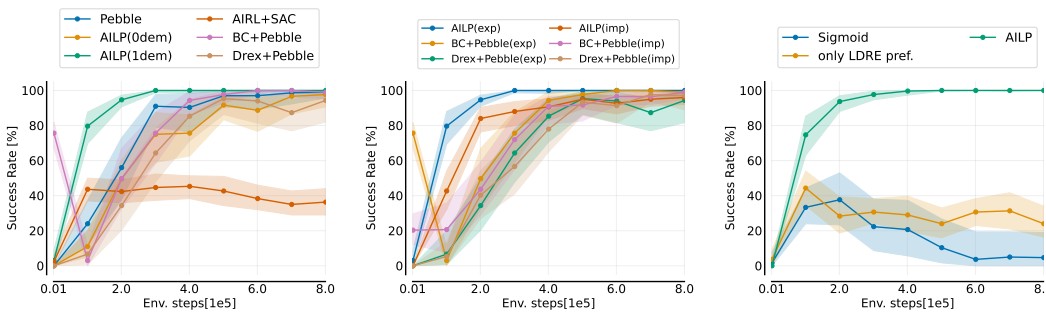

(a) Comparison of AILP and various baselines.  (b) Comparisons when using expert and imperfect demonstrations.  (c) Comparisons of AILP with different preference loss functions.

Figure 2: Interquartile mean for aggregated success rates for 6 metaworld tasks. Compared is our proposed method AILP with relative baselines and performance of AILP with modified preference loss function.

To answer **Q1**, we have compared our method to the Pebble algorithm (Lee et al., 2021a). We use the official implementation which is contained in the same code repository[1] as for (Lee et al., 2021b). For all evaluation we have used the parameters listed by (Lee et al., 2021b). In all our evaluations and for all methods we use the same network architecture for the actor and the critic of SAC, details are in the Appendix C. However, we use different networks for the reward function in Pebble and for the LDRE in AILP. Namely, for Pebble we use the network size from the the paper that consists of 3 layers with 256 nodes each. For AILP we use a smaller network with 2 layers with 64 nodes which is the best performing network architecture for AIL (Orsini et al., 2021). Pebble uses an ensemble of 3 reward networks which is motivated by increasing the stability of reward learning. However, we found no benefit from having an ensemble of LDRE networks in our method, and due to the increased computation time, we decided to train only a single network.

In order to answer **Q2** we add one expert demonstration to the start of learning process. Afterwards, in our experiments we do not add any demonstrations. Additionally, we also included the case of using our algorithm without any preference, in which AILP is practically a combination of AIRL and SAC[2]. Moreover, Orsini et al. (2021) show that the optimally performing AIL algorithm is also based on the combination of those two methods, and therefore it is the best option for AIL baseline in our paper. We can see in Figure 2 that AILP is successful in learning every task with the combined feedback. Using demonstrations helps us significantly in the beginning, thus increase the speed of the whole learning process. Moreover, we also tend to learn much faster in comparison to other baselines. Without preferences (AIRL+SAC) learning saturates early and success rates remain low. Additionally, we compared our method with two extensions of Pebble that consider demonstrations. In one case we used demonstrations to initialize the model policy using behavior cloning and start with preference learning using Pebble from that point. This approach is inspired by the work of Ibarz et al. (2018) but we also rely on the improvements provided in Pebble such as using SAC to optimize the reward. Note that in this case we do not have the entropy-based pretraining. Moreover, we also consider a combination of DREX and Pebble, in which we use the approach from DREX to generate the initial set of preferences, and then we apply Pebble from that point onward. We refer to this method as *DREX+Pebble*. In addition to the success rate presented in Figure 2, we report the values of normalized trajectories returns in Appendix D.1.

Additionally, in Appendix D.4, we present similar evaluations on a more difficult metaworld task, lever pull. Furthermore, we also evaluate the performance in a Mujoco task, HalfCheetah (Todorov et al., 2012). Details and results are in the Appendix D.5.

After the initial pretraining of $\tilde{\pi}$ we start adding 20 preferences each 5000 environmental steps, which equates to 20 full trajectories. In the case of Pebble the reward is updated only when we add new preferences. However, in the case of AILP because the LDRE depends on the current policy we need to update it more often, and we do that every 20 environmental steps. The updates are computationally expensive, especially since we increase the amount of preferences during training. Therefore, we limit the amount of previous preferences to 300 during training. We assume that the remaining, more recent preferences provide relevant information to guide the learning. Considering that the early preferences are sampled from older policies their contribution to learning is questionable if any. Moreover, given the frequent updates of the reward function training on fewer preferences would benefit us from the point of compute time. Entropy-based pretraining is performed for all baselines that are not initialized using BC and it is done once for each random seed and then shared for all experiments. Note that the performance of the algorithm from this initial part is not shown nor is counted in the steps shown in Figure 2. Moreover, in our comparisons we utilize interquartile means (Agarwal et al., 2021) over the 6 aggregated metaworld tasks as a more robust statistical measure.

For answering **Q3** we use partially trained policies to generate imperfect demonstrations. None of the policies was on average capable of successfully finishing the task. Using those demonstrations we ran the same set of experiments as before. The results are shown in Figure 5b. AILP learns the task successfully even in the presence of imperfect demonstrations and still benefits from the demonstration in comparison to the case with 0 demonstrations.

---

[1] https://github.com/pokaxpoka/B_Pref

[2] Here, we used a standard discriminator instead of the policy dependent discriminator architecture from the paper Fu et al. (2018). We were not successful using the original discriminator architecture together with SAC, yielding poor performance. Note that in the AIRL original paper, TRPO was used to train the policy.

**Ablation of the Preference Loss.** Finally, we present an ablation study regarding the used preference loss. Here, we ablate the sigmoid preference loss function used by Pebble (with using the log policy densities as additional rewards) as well as our algorithm using the hinge loss without the log policy density as reward. The results are shown in Figure 5c. The sigmoid loss enforces that the rewards of two trajectories in a preference pair are distinct, which causes instabilities in particular if the quality of the trajectories is quite similar. Here, the hinge loss shows a much more robust behavior and the agent's performance converges gracefully. Not using the log policy densities as additional rewards and hence treating the LDRE on its own as a reward function for preferences also leads to a poor performance as the LDRE is not a stationary reward function.

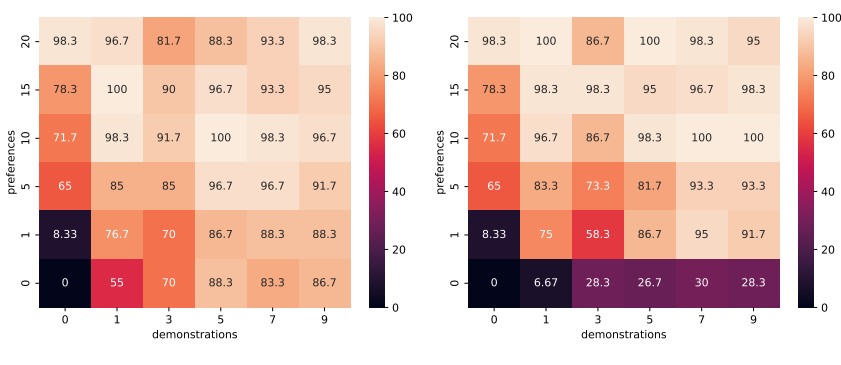

(a) Using expert demonstrations  (b) Using imperfect demonstrations

Figure 3: Success rates for AILP with different numbers of demonstrations and preferences for the window open task with expert and imperfect demonstrations. Adding additional demonstrations and preferences leads to orthogonal improvements in performance.

**Evaluating the Trade-off between Number of Preferences and Demonstrations.** We evaluate the performance of AILP for different amounts of preference and demonstrations on the *window open* task in Figure 3. We find that additional demonstrations generally improve performance, particularly for the case with a low number of preference. In the aforementioned figure we indicated the the number of preference that are added after 5 000 environmental steps. In the same figure we can compare the cases when we have expert and imperfect demonstrations. It can be observed that if we use more demonstrations, the number of preferences can be reduced in order to achieve a good performance. The contribution of additional demonstrations to the algorithm performance is most evident when we have a smaller number of preferences. Additional results over all 6 tasks with different number of demonstrations are provided in Appendix D.2.

## 6 CONCLUSION AND FUTURE WORK

We introduce Adversarial Imitation Learning with Preference, a novel method for learning from demonstrations and preferences. We modify and extend well-known Adversarial Imitation Learning methods to also be able to learn from preferences. This is achieved by using a log density ratio estimator, which is trained on separate losses for demonstrations and preferences. After training, the resulting network is used to express a reward function, and then used to optimize the generator policy. This procedure is iterated over until convergence and allows for novel teacher feedback in each loop. We show on a suite of simulated robotic manipulation tasks that this approach to learning from preferences benefits from multiple types of feedback, outperforming the state of the art.

A limitation of our proposed method is the number of environment steps needed when updating the generator policy which is challenging on a real robot. In future research, we will investigate the use of multiple trajectories and ask for *rankings* from the teacher (Myers et al., 2021). Additionally, we will expand AILP to deal with preference queries where two options are too similar for teachers to compare and we will also evaluate our approach with human-generated data.

ACKNOWLEDGMENTS

GN was supported by the Carl Zeiss Foundation under the project JuBot (Jung Bleiben mit Robotern). NF was supported by the BMBF project Davis (Datengetriebene Vernetzung für die ingenieurtechnische Simulation).

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

## A  METAWORLD ENVIRONMENTS

All tasks in the metaworld benchmark (Yu et al., 2019) share the same observation and action space. The observation space is 39 dimensional and consists of the of the current and previous state and a desired position, with the last element related to the position of the goal object, such as the button in the button press scenario. Each state has 18 dimensions and consists of the end-effector position (3 dimensions), distance between the two fingers of the gripper (1 dimension), Euclidean position (3 dimensions) and orientation expressed using 4 dimensional quaternions of object 1 and the same for an object 2. Amongst considered tasks we only have environments with a single object, and the values of the position and orientation are set to zero. This object is defined by the task, e.g., in the button press is the button. The action space is defined by controls of the position of the end-effector as well as the distance between gripper fingers. The orientation of the end-effector is fixed such that it always points towards the table. Using environmental rewards and SAC we have trained expert robot policies and the mean and standard deviation of trajectory returns over 100 rollouts are presented in Table 1.

| Task | Expert | Imperfect |
|------|--------|-----------|
| Button press | 1773.0 | 614.7 |
| Door open | 2015.9 | 769.9 |
| Window open | 1982.9 | 681.3 |
| Door close | 2024.0 | 1166.4 |
| Drawer close | 2032.2 | 1473 |
| Window close | 2032.2 | 395.5 |

Table 1: Mean of trajectory returns over 100 rollouts from an expert and imperfect policy.

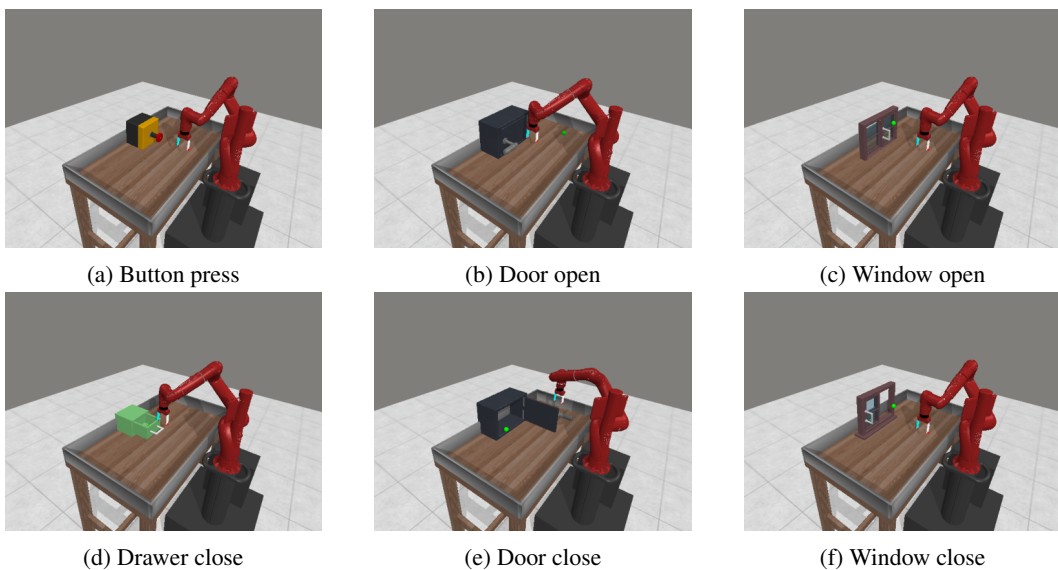

(a) Button press        (b) Door open        (c) Window open

(d) Drawer close        (e) Door close        (f) Window close

Figure 4: Manipulation tasks from the metaworld benchmarking task set (Yu et al., 2019). (a) In the button press task, the robot needs to reach and press the button. (b) In the door open task, the robot needs to grab the handle of the door and move it to the green point, thus opening the door. (c) In window open task, the robot needs to move the handle to the left edge of the window. The button, the door, the drawer, and the window are randomly placed in front of the robot for each rollout. (d) In the drawer close task, the robot needs to close the drawer. (e) In the door close task, the robot needs to move the door to the green point, thus closing the door. (f) In window close task, the robot needs to close window. The button, the door, the drawer, and the window are randomly placed in front of the robot for each rollout.

## B    FULL ALGORITHM

The AILP algorithm with all individual steps in shown in Alg.1 below.

**Data:** demonstrations $\mathcal{D} = \{\xi_i\}$, preferences $\mathcal{P} = \{\xi_{i,1}, \xi_{i,2}\}$
bootstrap initial policy $\tilde{\pi}_0(s)$ to maximize state entropy $H(s)$
$i = 0$
**while** *i < max iterations* **do**
    Record demonstrations and preferences, update $\mathcal{D}$ and $\mathcal{P}$
    Generate samples from current policy $\tilde{\pi}_i(\cdot|s)$ and add to replay buffer B
    **Step 1: Update LDRE $\phi_i$**
    Get model samples $\mathcal{B}_i$ from the replay buffer B
    Optimize demonstration loss $L_{dem}(\phi_i, \mathcal{D}, \mathcal{M}_i)$
    Optimize preference loss $L_{pref}(R_i(\phi_i), \mathcal{P})$
    **Step 2: Update policy**
    Relabel buffer B using $-\phi(s,a)$
    Using SAC with $-\phi(s,a)$ as reward, update $\tilde{\pi}_{i+1}$
    i = i+1
**end**

**Algorithm 1:** Adversarial Imitation Learning with Preferences. In each iteration we generate samples from the current policy $\tilde{\pi}_i(a|s)$, then we optimize the demonstration and preference losses to update our LDRE $\phi(s,a)$. Optionally, depending on the iteration we generate new preference samples or add demonstrations. After we have updated LDRE $\phi(s,a)$ we use it to define a reward function $R(s,a)$ that is then optimized using SAC. Using the updated policy we generate new samples and reiterate over the whole process.

## C    SAC HYPERPARAMETERS

In all evaluated experiments in Section 5 we use the same parameters for SAC and those are listed in Table 2.

| Hyperparameter | Value |
|---|---|
| Learning rate | 0.0003 |
| Optimizer | Adam |
| Critic target update frequency | 2 |
| Discount factor $\gamma$ | 0.99 |
| Batch size | 512 |
| Initial temperature | 0.1 |
| $(\beta_1, \beta_2)$ | (0.9,0.999) |
| Actor network (nodes per hidden layer) | [256,256,256] |
| Critic network (nodes per hidden layer) | [256,256,256] |
| Critic EMA $\tau$ | 0.005 |

Table 2: Hyperparameters for SAC that are shared for all experiments

## D    ADDITIONAL EXPERIMENTS AND ABLATIONS

### D.1    NORMALIZED TRAJECTORY RETURNS

In addition to the success rate that is presented in Figure 2, we have also analyzed the normalized trajectory returns for the 6 metaworld tasks. Those results are presented in Figure 5. We observe same trends and relative performance between our algorithm and the baselines as we see in success rate results.

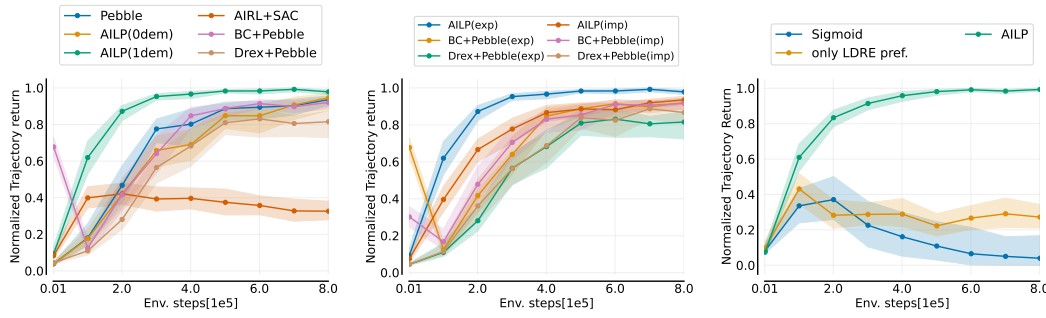

(a) Comparison of AILP and various baselines.

(b) Comparisons when using expert and imperfect demonstrations.

(c) Comparisons of AILP with different preference loss functions.

Figure 5: Interquartile mean for aggregated normalized trajectory returns for 6 metaworld tasks. Compared is our proposed method AILP with relative baselines and performance of AILP with modified preference loss function.

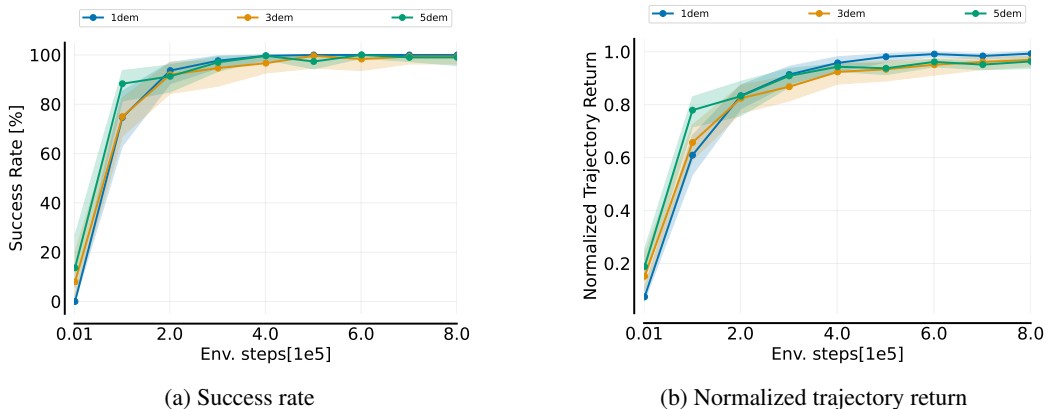

(a) Success rate

(b) Normalized trajectory return

Figure 6: Direct comparison of AILP with more demonstrations

## D.2 DIRECT COMPARISON OF USING MORE DEMONSTRATIONS

For the default setup from Section 5, we also run an ablation with additional demonstrations over all 6 tasks. Results are shown in the Figure 6:

## D.3 INFLUENCE OF WEIGHTED PREFERENCE LOSS

We have evaluated our method when trained with the preference loss function $L_{pref}(R_i(\phi_i), \mathcal{P})$ that is scaled by a hyperparameter $\alpha$. In Figure 7, we show the performance of our algorithm for values of $\alpha \in [0.1, 1, 10, 100]$. Additionally, we include two extreme cases, AILP with no demonstrations and AIRL+SAC that theoretically corresponds to the case of $\alpha = 0$. We observe that for $\alpha = 0.1$, our method performs similar to the case without preferences (AIRL+SAC). Likewise, for $\alpha = 100$, AILP performs similar to the case when we have only preferences.

In Figure 8 we present the performance of our method with imperfect demonstrations. Similar conclusion can be drawn about the influence of $\alpha$ on the performance of our method as in the case with expert demonstrations. As we increase the value of $\alpha$, we rely in the learning process more on preference data.

## D.4 LEVER-PULL METAWORLD TASK

Besides the 6 metaworld tasks in Appendix A, we additionally evaluate lever pull task. This task is significantly harder, and our expert policy which with obtained by training an agent using SAC

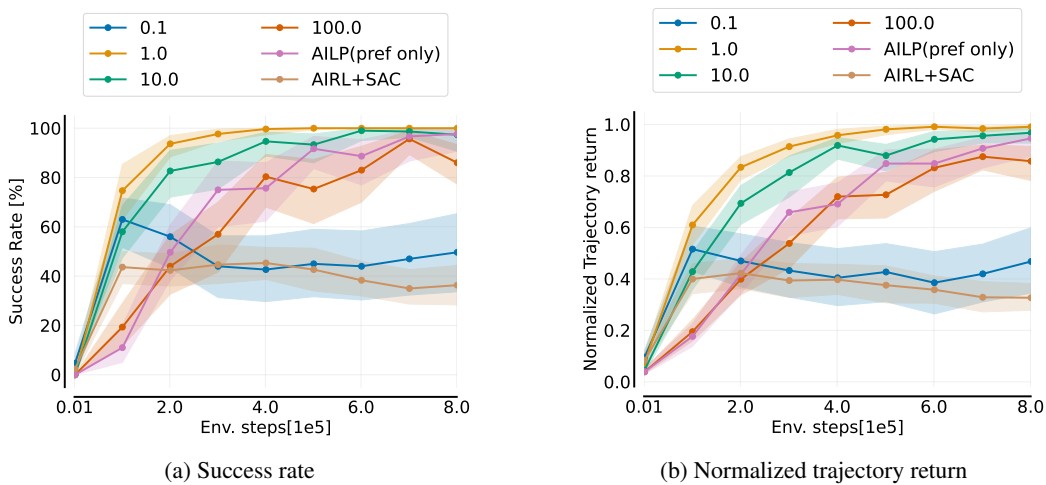

(a) Success rate

(b) Normalized trajectory return

Figure 7: Comparison of different values of $\alpha$ when optimizing $\alpha P_{pref}$ with expert demonstrations.

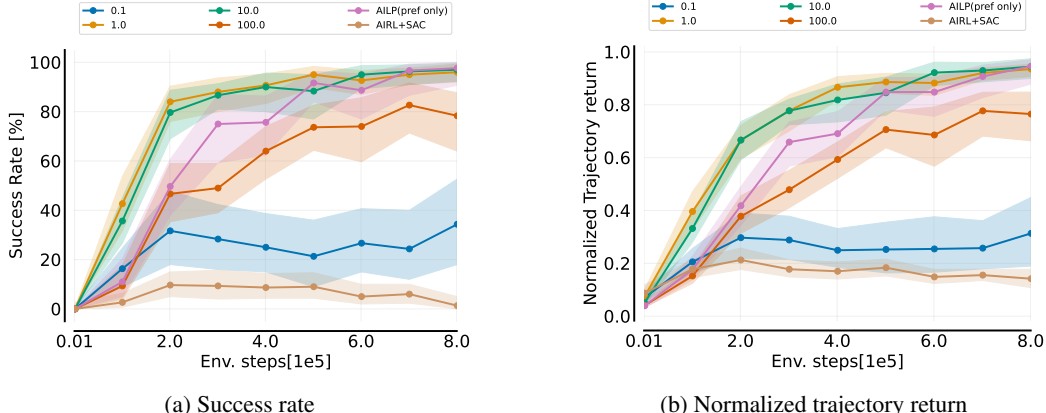

(a) Success rate

(b) Normalized trajectory return

Figure 8: Comparison of different values of $\alpha$ when optimizing $\alpha P_{pref}$ with imperfect demonstrations.

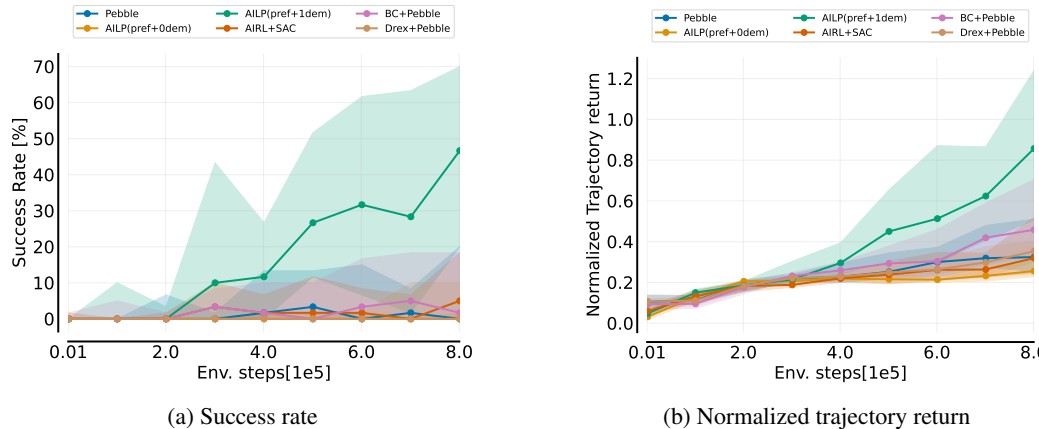

(a) Success rate            (b) Normalized trajectory return

Figure 9: Evaluation of the Lever-pull task

with the environmental reward, was only capable of successfully finishing the task in $22\%$ of the evaluated rollouts. In Figure 9 are presented the success rate and normalized trajectory returns for our method as well as for the same baseline we evaluated in Section 5. We notice that all algorithms struggle with this task, but after some time, our method starts to learn and noticeably improve its performance. Still the trained policy on average has not reached the $50\%$ success rate, and the variance is noticeable, but on average it is performing better than the expert we used to generate the demonstrations.

### D.5 HALFCHEETAH TASK

HalfCheetah is a common reinforcement leaning benchmark locomotion task (Todorov et al., 2012). In comparison to metaworld tasks we can claim that it is a simpler task, because for each metaworld task, the initial position of a target object, such as the button in the button press task, is moved, but for the HalfCheetah we don't have such substantial changes. Additionally, it's a nonepisodic. In Figure 10, the normalized trajectory returns of our method and the set of baselines we evaluated in Section 5 are shown for this task. We can notice that initially Pebble and BC+Pebble are performing better, but our method achieves similar performance at the end of the training. Moreover, it is worth noting that in AIL papers, tasks are usually made more difficult by sub-sampling demonstrations so only every 20th step is considered. Under these circumstances, AIL methods perform better. However, for a completely fair comparison in this paper we have not sub-sampled expert demonstrations. To normalize the trajectory returns we use the average return of our expert policy.

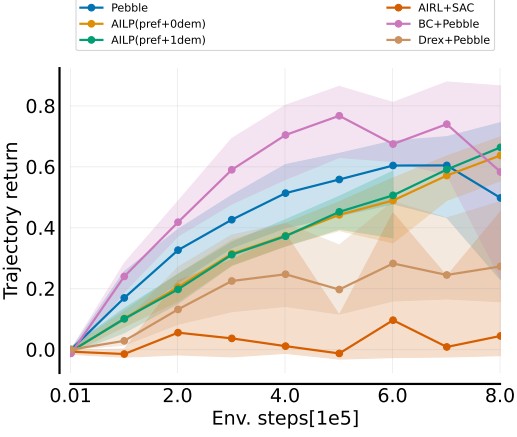

Figure 10: Normalized trajectory return of various algorithm for the HalfCheetah task

