# OpenReview forum: "Adversarial Imitation Learning with Preferences"
_ICLR.cc/2023/Conference — ICLR 2023 poster_

### Official Review · Reviewer_suaS · 2022-10-24

**Confidence:** 4
**Correctness:** 4
**Technical Novelty And Significance:** 3
**Empirical Novelty And Significance:** 3
**Recommendation:** 6

**Clarity, Quality, Novelty And Reproducibility:**

The paper presentation is clear and well written. While parts of the method is based on prior work, the proposed method and means of combining preferences and demonstrations seem novel. Code is not provided so reproducibility is unclear.

**Strength And Weaknesses:**

Strengths
- The motivation is compelling and the paper is well written, easy to follow. In particular, the use of a density ratio estimator for both AIL and for judging preferences seems reasonable.
- The empirical results seem compelling, and compared against relevant baselines (of AIL methods, preference-based methods, and naive combinations of the two). The experimental results also present with ablations on their reward formulation choice and compare the effect of providing varying numbers of demonstrations and preferences. In particular, the combination of preferences and imperfect demonstrations is an interesting setting — although in this work it does not seem to show a big improvement over just using the preferences.

Weaknesses
- The main paper of comparison (PEBBLE) shows experiments across a wider set of environments (including locomotion tasks like Quadruped, Walker, etc.) and using real human preferences. It would be an interesting and stronger comparison if those same experiments were replicated here as well, with AILP.
- It also seems like in these domains, a single demonstration is sufficient to saturate performance. It would be interesting to find a domain where more demonstrations are actually helpful / necessary.
- Formatting could be improved. Plots are currently a little hard to read, should increase legend size or move them off the figures.

Questions
- Is relabelling the data in the buffer with the updated reward function time consuming and/or difficult to do as the buffer grows? How frequently is this relabelling necessary?
- This method assumes preferences are aligned with expert demonstrations. I would be curious to see what happens if preferences weren’t exactly aligned -- e.g. if the preferences were trying to guide for more stylistic behaviours.

**Summary Of The Paper:**

This work proposes policy training that combines two commonly used types of human feedback — demonstrations and preferences, each of which has its own strengths and weaknesses. Specifically, the authors propose a method for incorporating preferences into adversarial imitation learning as a unified framework, by reformulating adversarial imitation learning with density ratio estimation, then using the same learned density ratio estimator to learn from preferences. The authors compare their method against other preference based methods and adversarial IL methods, as well as naive combinations of imitation learning and preference based learning.

**Summary Of The Review:**

I recommend a 6. The overall methodology seems sound and experimental results empirically support the author’s claim that AILP manages to combine learning from both demonstrations and preferences. The paper could be improved with more direct comparisons to PEBBLE (more diverse set of tasks, seeing if the method still works when learning from human preferences for stylistic behaviours, etc.).

---

> ### Author Response · Authors · 2022-11-15
> **Answer to Reviewer suaS**
>
> We thank the reviewer for their constructive comments and suggestions.
>
> Currently we are running additional experiments in more difficult metaworld environments, as well as in some Mujoco locomotion tasks. We will include all the results in the updated version of our paper that we will upload before Friday, 18th of November. In this moment we would like to address other points raised by the reviewer and answer their question.
>
> The formatting of the plots has been improved according to the suggestions and we moved the legend to the side. We thank the reviewer for the advice. Furthermore, we will make the code publicly available when the paper gets accepted.
>
> Regarding the question about relabelling of the buffer; instead of relabelling the entire buffer itself, we dynamically re-label the data points that are used for each mini-batch during policy training with SAC. This is done before each policy update step, and it requires only an additional forward pass through the discriminator, resulting in a minor increase in computation time when compared to the full training step.

---

> ### Author Response · Authors · 2022-11-18
> **Uploaded updated Paper**
>
> We would like to thank again the reviewer for their comments and feedback. The updated version of our paper is now uploaded and the parts that specifically address concerns and questions that the Reviewer suaS raised are marked in orange.
>
> Specifically:
> - In Figure 3 besides the case with expert demonstrations for the window open task, we have also added the case with imperfect demonstrations. With these 2 figures, we also want to address the raised issue that one demonstration is enough to saturate the performance, as it can be observed that with access to less preferences, we would need more demonstrations.
> - Evaluation of an additional, more difficult metaworld task, in which our expert has success rate of 22%. The task was not solved with any of the evaluated methods, but our method was able to obtain results better than other baselines as well as better average success rate than the expert.
> - Evaluation of a Mujoco locomotion task. We have evaluated the HalfCheetah in the same setup as the metaworld task. The performance of our method at the end of the training is comparable to the baselines.
> - Improved formatting of Figure 2. Moreover, we have moved part of the results to the Appendix.
>
> Additionally, we would like to note that we have run a preliminary experiment with the Mujoco HalfCheetah with preferences obtained from a human in which we tried to train it run backwards. Initial tests were successful and show that it is possible to learn with our method from human provided preferences. However, due to time constraints we are not able to carry out a complete study and a comparison with the baselines, and therefore, we will do it as a part of future work.

---

### Official Review · Reviewer_qpWS · 2022-10-25

**Confidence:** 3
**Correctness:** 3
**Technical Novelty And Significance:** 2
**Empirical Novelty And Significance:** 2
**Recommendation:** 5

**Clarity, Quality, Novelty And Reproducibility:**

I think the paper is overall well-written, while the problem setup is not clearly discussed in Section 3. Even though the introduction of adversarial IL is thorough, it is not clearly described whether all trajectories generated by the learner can be evaluated by the expert to generate preferences, or the preferences are obtained via some query strategy. I suggest describing the problem setup, in special how the preferences are obtained, clearer in the paper.


**Strength And Weaknesses:**

Strengths:

1. The experiments show significant performance improvement over existing baselines under the LfD + LfP setting.

2. In my view, the paper is technically sound. The proposed approach is reasonable to work in practice.

Weaknesses:

1. I think the technical contribution is somehow limited. Comparing to existing adversarial IL approaches, the essential part of the proposed method is in Equation (9), which is a ranking loss to make use of the preference information. However, the hinge loss is one of the common choices for ranking tasks.

2. From the paper, I am not quite clear why the proposed method can achieve such good performance over the baselines, since the algorithm itself does not show significant novel improvements. I also suggest including more discussions in the paper.

3. I think it would be useful if the code could released when the paper gets accepted.

**Summary Of The Paper:**

In this paper, the setting of the combination of learning from demonstration (LfD) and learning from preference (LfP) is studied. Under the adversarial imitation learning (IL) framework, a new algorithm is proposed, whose main idea is to include a new ranking loss term for LfP to combine with the adversarial LfD loss. The proposed method is tested under the meta-world benchmark tasks, showing performance improvement over existing baselines.



**Summary Of The Review:**

In summary, I think this is a solid paper proposing an empirically well-performed algorithm. However, I think the technical contribution is relatively weak, which is the major concern to me.

---

> ### Author Response · Authors · 2022-11-15
> **Answer to Reviewer qpWS**
>
> We thank the reviewer for their constructive comments and detailed feedback. In the following, we want to address the individual concerns mentioned by the reviewer.
>
> Regarding the first raised issue of missing novelty, we respectfully disagree with the reviewer that there is hardly any novelty and want to point out that they might have missed important aspects of the paper which also constitute to the main part of our contribution. We will therefore also clarify the contribution more clearly in the paper to avoid such confusions. Our main contribution is not introducing the hinge loss in Equation (9) but how we can integrate preferences and demonstrations in an adversarial imitation learning setting. To do so, we have mathematically derived a novel reward function that depends also on the density of the current policy (see Equation 7). Using this definition of the reward enables us to express a preference loss function (that typically learns a reward) that is compatible with the discriminator loss (that does *not* learn a reward) from AIL. This is the major contribution of the paper, not Equation 9. Our ablation studies also confirm this by showing that without including the log density of the policy as reward, the performance degrades, (see Figure 2c) emphasizing the importance of this contribution. Regarding the hinge loss, we agree that the hinge loss has been used before (although not so much in recent preference learning work using DNNs) and we do not claim this is a main contribution. Yet, we are not aware of a proper comparison of sigmoid and hinge loss for preferences using DNNs, so using the hinge loss in this setup and showing that it is competitive also represents a small, but valuable contribution. We will clarify this also in the paper. We hope with this explanation we also addressed the question of why our performance is better than the pure preference-based (PEEBLE) or pure demonstration-based (AIRL+SAC) baselines - We use a combination of demonstrations and preferences; hence, our approach can exploit more information which leads to faster learning. In order to be able to use this combination, our main contribution (novel reward definition in AIL) is required as shown by our ablations.
>
> Furthermore, we have now included in the paper an additional description about the problem setup, and specifically about preference generation process, entropy-based sampling [1], that was not described in the initial paper due to paper length. We will upload the updated version of the paper by Friday, the 18th of November. Moreover, we will make the code publicly available when the paper gets accepted.
>
> We are looking forward to further discussion about any additional or remaining questions and concerns.
>
> [1] Kimin Lee, Laura Smith, and Pieter Abbeel. Pebble: Feedback-efficient interactive reinforcement learning via relabeling experience and unsupervised pre-training. International Conference on Machine Learning, 2021

---

> ### Author Response · Authors · 2022-11-18
> **Uploaded updated Paper**
>
> We would like to thank again the reviewer for their comments and feedback. The updated version of our paper is now uploaded and the parts that specifically address concerns that the Reviewer qpWS raised are marked in green color. Specifically:
>
> - Improved formulation of our contribution.
> - Additional description of the preference generation process.

---

### Official Review · Reviewer_nvkH · 2022-10-29

**Confidence:** 3
**Correctness:** 3
**Technical Novelty And Significance:** 3
**Empirical Novelty And Significance:** 3
**Recommendation:** 6

**Clarity, Quality, Novelty And Reproducibility:**

AILP is a novel and interesting algorithm. In addition, The pseudocode provided in the appendix provides a clear understanding.

**Details Of Ethics Concerns:**

-

**Strength And Weaknesses:**

**Strength**

AILP successfully handles feedback from demonstrations and preferences effectively.

**Weakness**

- AILP uses two losses, $L_\text{dem}(\phi_i,\mathcal{D},\mathcal{M}_i)$ and $L_\text{pref}(R_i(\phi_i \mathcal{P})$ to update LDRE $\phi_i$. However, I am wondering if a stable update of $\phi_i$ is possible using both losses.
- In Eq. (10), The authors remove the KL term, but the entropy term is added due to the usage of SAC. Consequently, $\sum_{t=1}^T\mathbb{E}_{p^\pi(s_t)}[-\log\pi_k(\cdot|s_t)]$ is removed. Does it make sense to remove this term without a theoretical analysis?

**Questions**

- I'm just wondering what's the problem with using a loss like $L_\text{dem}(\phi_i,\mathcal{D},\mathcal{M}_i)+ \alpha L_\text{pref}(R_i(\phi_i), \mathcal{P})$? Here, $\alpha\in[0,1]$ is a hyperparameter.
- Why is there no AIL algorithm as a baseline for experimental evaluation? In addition, do baselines also use the pre-trained policy $\tilde\pi$ to maximize state entropy?


**Summary Of The Paper:**

This paper proposes a novel algorithm, named adversarial imitation learning with preferences (AILP), for learning from demonstrations and preferences. AILP builds upon the well-known adversarial imitation learning (AIL) framework which uses discriminator to construct reward function. To train the discriminator, AILP uses a combination of a BCE classification loss and a hinge loss per preference pair. Then, AILP uses SAC to update policy where the reward term is composed of the trained discriminator. Finally, the authors provide empirical comparison of AILP and baselines on the 6 different manipulation tasks from the meta-world benchmark.

**Summary Of The Review:**

This paper proposes an interesting policy learning algorithm which uses both demonstrations and preferences. However, I have some questions about this algorithm.

---

> ### Author Response · Authors · 2022-11-15
> **Answer to Reviewer nvkH**
>
> We thank the reviewer for their constructive comments, and detailed feedback, and especially for the positive review. In the following, we want to address the individual points mentioned by the reviewer. We are currently running additional experiments that address points that the reviewer has made, and we will include them in the updated version of the paper that we will upload by Friday, the 18th of November.
>
> Specifically, we thank the reviewer for the observation made about the stability of the update of $\phi$ when using both $L_{dem}$ and $L_{pref}$. We currently optimize the losses consequently for a practical reason. Namely, because we are dealing with two different data types, each is divided into different batch sizes. From the theoretical view there is no issue in terms of stability as we just add two loss functions that we want to minimize jointly using gradient descent.
>
> In order to address this point and also answer the question about using a loss like $L_\text{dem}(\phi_i,\mathcal{D},\mathcal{M}_i)+ \alpha L_\text{pref}(R_i(\phi_i), \mathcal{P})$, we are running additional experiments that include evaluation with different values of $\alpha$, as proposed in the review. Additionally, we would like to add that the hyperparameter $\alpha$ could be larger than 1. This corresponds to the case when we would like to rely more on the preference data, and initial results support this claim.
>
> Regarding the second raised issue about removing the KL term in Equation 10, we would like to note that the form of policy optimization in our paper corresponds to the optimization done by AIRL [1] in the manner that is explained in [2], albeit they express the reward using a discriminator instead of a log density ratio estimator. We have now expanded Subsection 4.4 to include these additional clarifications. Regarding the question about adversarial imitation learning baselines, we would like to state that we have used AIRL+SAC as an AIL baseline. AIRL[1] is an adversarial imitation learning baseline, but in comparison to the original work, we used SAC to optimize it because recent work has shown optimal performance [2]. We would also like to mention that in comparison to the original AIRL paper, we do not estimate the reward function in order to provide a fair comparison with the baselines. Regarding the second part of this question about the baseline initialization, we use pre-trained policy $\tilde{\pi}$ to initialize all baselines except those that use BC.
>
>
>
> [1] Justin Fu, Katie Luo, and Sergey Levine. Learning robust rewards with adversarial inverse reinforcement learning. Sixth International Conference on Learning Representations ICLR, 2018.
>
> [2] Manu Orsini, Anton Raichuk, Leonard Hussenot, Damien Vincent, Robert Dadashi, Sertan Girgin, Matthieu Geist, Olivier Bachem, Olivier Pietquin, and Marcin Andrychowicz. What matters for adversarial imitation learning? Advances in Neural Information Processing Systems, 6 2021.

---

> ### Author Response · Authors · 2022-11-18
> **Uploaded updated Paper**
>
> We would like to thank again the reviewer for their comments and feedback. The updated version of our paper is now uploaded and the parts that specifically address concerns and questions that the Reviewer nvkH raised are marked in blue color. Specifically:
>
> - Ablation study with weighted preference loss $\alpha L_{pref}(R_i(ϕ_i),P)$. We evaluated our method with different values of $\alpha$ for the case with expert demonstrations as well as imperfect ones. Results show that with small $\alpha=0.1$, we rely more on demonstrations, while higher values lead to $\alpha=100$ suboptimal initial performance to over relying on preferences which are initially less informative. Besides the additions to the main paper, we have dedicated a section in the appendix to show the results with imperfect demonstrations.
> - Additional support for the assumption to remove the KL term in Subsection 4.4 Optimizing the Generator.
> - Clarification about using AIRL+SAC as an AIL baseline that has been shown to have the best performance among AIL baselines.
> - Clarification about using entropy-based pretraining for baselines.

---

### Author Response · Authors · 2022-11-18
**Uploaded updated Paper**

Dear Area chair and Reviewers,

attached is the revised version of our paper. For easy reference, we have color-coded all changes depending on the reviewer who suggested them:

- Reviewer ***nvkH***: blue.
- Reviewer ***qpWS***: green.
- Reviewer ***suaS***: orange.

Additionally, we briefly list the changes made to the paper, and in the parenthesis, we denote the reviewer whose comments motivated these changes,

- Ablation study with weighted preference loss $\alpha L_{pref}(R_i(ϕ_i),P)$.  (nvkH)
- Additional support for the assumption to remove the KL term in Subsection 4.4 Optimizing the Generator. (nvkH)
- Clarification about using AIRL+SAC as an AIL baseline that has been shown to perform optimally. (nvkH)
- Clarification about using entropy-based pretraining for baselines. (nvkH)
- Improved formulation of our contribution. (qpWS)
- Additional description of the preference generation process. (qpWS)
- Additional experiments with imperfect demonstrations for the window open task in Figure 3. (suaS)
- Evaluation of an additional, more difficult metaworld task. (suaS)
- Evaluation of a Mujoco locomotion task. (suaS)
- Improved formatting of Figure 2. (suaS)

We would like to thank the reviewers for their helpful and insightful comments. We look forward to hearing your opinion about the updated version of the paper. We hope that we were able to address all of the raised concerns and answer all questions. In the case, that any concern is left unaddressed by our replies and the revised paper, please let us know.

Best regards,

Authors

---

### Decision · Program_Chairs · 2023-01-20

**Decision:**

Accept: poster

**Justification For Why Not Higher Score:**

Problem is well-needed, but their proposed approach has limited technical contribution.

**Justification For Why Not Lower Score:**

* Clear problem formulation / motivation / novelty
* Thorough experimental results

**Metareview: Summary, Strengths And Weaknesses:**

This paper proposes a method that can utilize both preference and demonstration from an expert. While, as one reviewer concerned about, the technical novelty is not high (simply combining two existing methods), the problem formulation is novel and it is well motivated in the paper. I am convinced that this is a needed setup. Also, the paper provided a good set experiments, adding extra experiments reviewers requested, with all positive results. It is a good paper to share with the community.

**Note From Pc:**

if the above contains the word "oral" or "spotlight" please see: "oral" presentation means -> notable-top-5% and "spotlight" means -> notable-top-25%. As stated in our emails, we are disassociating presentation type from AC recommendations